# An insect anti-antiaphrodisiac

## Colin S Brent[1]*, John A Byers[1,2], Anat Levi-Zada[3]

[1]Arid Land Agricultural Research Center, United States Department of Agriculture-Agricultural Research Service, Maricopa, United States; [2]Department of Entomology, Robert H. Smith Faculty of Agriculture, Food and Environment, The Hebrew University of Jerusalem, Rehovot, Israel; [3]Department of Entomology-Chemistry Unit, Agricultural Research Organization, Volcani Center, Rishon LeZion, Israel

**Abstract** Passive mechanisms of mate guarding are used by males to promote sperm precedence with little cost, but these tactics can be disadvantageous for their mates and other males. Mated females of the plant bug *Lygus hesperus* are rendered temporarily unattractive by seminal fluids containing myristyl acetate and geranylgeranyl acetate. These antiaphrodisiac pheromones are gradually released from the female's gonopore, declining until they no longer suppress male courtship. Because starting quantities of these compounds can vary widely, the repellant signal becomes less reliable over time. Evidence was found of a complimentary mechanism that more accurately conveys female mating status. Once inside the female, geranylgeranyl acetate is progressively converted to geranylgeraniol then externalized. Geranylgeraniol counteracts the antiaphrodisiac effect despite having no inherent attractant properties of its own. This is the first evidence for such an anti-antiaphrodisiac pheromone, adding a new element to the communication mechanisms regulating reproductive behaviors.

*For correspondence: colin.brent@ars.usda.gov

**Competing interests:** The authors declare that no competing interests exist.

## Introduction

Chemical signaling is an essential part of the regulation of mating in many insects, with a combination of pheromonal attractants and repellents indicating the suitability of prospective mates (*Gillott, 2003*). Several species have been shown to rely upon the transfer of an antiaphrodisiac from male to female during mating, the effect of which is to reduce the sexual attractiveness of females concurrent with a post-copulatory ovipositional period (*Happ, 1969*; *Gilbert, 1976*; *Kukuk, 1985*; *Tompkins and Hall, 1981a*, *1981b*; *Jallon et al., 1981*; *Scott, 1986*; *Andersson et al., 2000*, *2003*; *Schulz et al., 2008*; *Yew et al., 2009*). The mating male benefits from a reduced risk of sperm competition, while potential successor suitors avoid sperm competition as well as reduce the energetic costs and predation risks associated with courting a female that is unlikely to be receptive (*Gillott, 2003*; *Malouines, 2016*). Females benefit from this change in their chemical signature by a reduction in male harassment, which might otherwise negatively impact longevity, ovipositional opportunities, and predation avoidance (*Forsberg and Wiklund, 1989*; *Magnhagen, 1991*; *Cook et al., 1994*; *Clutton-Brock and Langley, 1997*; *Bateman et al., 2006*; *den Hollander and Gwynne, 2009*). This system is particularly useful for species in which females mate only once and for whom a protracted or permanent loss of attractiveness has no negative consequences (*Gillott, 2003*). Similarly, mated females can also cease releasing their attractant pheromones (*Raina, 1989*; *Kingan et al., 1993*; *Ayasse et al., 1999*; *Eliyahu et al., 2003*; *Fukuyama et al., 2007*; *Oku and Yasuda, 2010*), potentially adding to the impact of an antiaphrodisiac. However, in species with females that can or need to mate multiple times over their lives, the antiaphrodisiac might actually fail to accurately convey a female's reproductive state to conspecific males (*Malouines, 2016*).

**eLife digest** In many animal species, males guard females to prevent rivals from mating so that they can be sure that they fathered the female's offspring. Some guarding methods work even when the male is not present. For example, the semen of some male insects contains chemicals known as antiaphrodisiacs that repel other males from females who have recently mated. Over the course of several days or weeks, the females expel or degrade the antiaphrodisiacs, making themselves attractive to other mates again. How long it takes to eliminate the antiaphrodisiacs depends on how much of the chemicals were deposited in the first place. Therefore, males could gain an advantage in fertilizing more eggs by depositing excess antiaphrodisiac to make the females unattractive to other mates for a long time.

The Western tarnished plant bug (*Lygus hesperus*) is an agricultural pest that targets cotton, strawberries and other crops. One antiaphrodisac had already been identified in the semen of male *Lygus* bugs. To investigate whether the males produced any others, Brent et al. tested the molecules emitted by recently mated females. This search identified another potential antiaphrodisiac. However, females are able to convert this second chemical into a third one that neither attracts nor repels males. This "anti-antiaphrodisiac" acts against the males' two antiaphrodisiacs, and allows the females to more accurately signal when they are ready to mate again, giving them more control over their reproduction.

Anti-antiaphrodisiacs were not previously known to exist, but now that scientists know where to look, more are likely to be found in other species. A better understanding of how different chemicals interact to influence the mating behavior of insects could also lead to new methods of targeting pests of crops, which are safer for the environment than existing chemical pesticides.

Often the antiaphrodisiac consists of just one or at most a few chemicals that are repellant to males (*Jallon et al., 1981*; *Andersson et al., 2000*; *Schulz et al., 2008*; *Yew et al., 2009*; *Zawistowski and Richmond, 1986*; *Krueger et al., 2016*) or that mask a female's attractants (*Andersson et al., 2003*; *Zhang and Aldrich, 2003*; *Zhang et al., 2007*). These pheromones are emitted from the female over days or weeks until fully discharged or degraded, at which point the female can attract a new mate. One disadvantage of such a simple signaling system is that the amount of antiaphrodisiac being emitted by a female may not coincide with her readiness to mate again. A male's maturity, health, or the interval between insemination events can all influence the amount of antiaphrodisiac that he can transfer along with his sperm. There is even evidence that males can intentionally bias the size of their spermatophore in response to female mating history and the local level of intrasexual competition (*Larsdotter-Mellström et al., 2016*). Such variability in the starting amount can result in a female being ready to mate well before a large load of antiaphrodisiac is sufficiently depleted for her to be attractive again, or being prematurely courted if the male transfers too little. Such signaling uncertainty is potentially costly to the fitness of both females and males, and should create selective pressure to produce a more accurate signaling system that incorporates information beyond the amount of remaining antiaphrodisiac (*Estrada et al., 2011*). To date, the only mechanism shown to allow females to counteract the effect of an antiaphrodisiac is in *Drosophila*, in which females actively eject mating-transferred cis-vaccenyl acetate from their reproductive tract (*Laturney and Billeter, 2016*).

Females of the western tarnished plant bug, *Lygus hesperus* Knight, are polyandrous, mating repeatedly throughout their lives to ensure a steady supply of sperm and to maintain an elevated rate of oviposition (*Strong et al., 1970*; *Brent, 2010a*; *Brent and Spurgeon, 2011*; *Brent et al., 2011*). Mating also causes these females to become less attractive than virgins to males (*Strong et al., 1970*; *Brent, 2010a*). A sex-pheromone has been identified for this species (*Byers et al., 2013*), but there is no evidence that the release rate of this chemical blend is affected by the mating status of a female. All evidence suggests that *L. hesperus* female attractiveness is instead modulated by myristyl acetate (MA), a volatile compound transferred during insemination that is released via the female's vaginal pore, where it is detected by antennating males and acts as an antiaphrodisiac (*Brent and Byers, 2011*). Mated females remain unattractive to males for a 4–5

day post-copulatory period during which they also become sexually unreceptive and increase their rate of oviposition (*Brent, 2010b*). MA is produced in the male accessory glands and transferred along with other components in the spermatophore during mating (*Brent and Byers, 2011*). Some of these other seminal components may also play a role in male assessment of female mating history during courtship. Although females regain their attractiveness to prospective mates over time (*Strong et al., 1970*; *Brent, 2010b*), the mechanism by which this is accomplished, either by passive degradation or active countermeasure, had not been determined.

Here, we undertook a detailed analysis of the composition of *L. hesperus* spermatophores and the compounds emitted by mated females to determine what other volatiles might shape male reproductive behavior and how the compounds change over time. We used gas-chromatography-mass spectrometry coupled with behavioral assays to identify compounds, in addition to MA, that are involved in modulating male responses to potential mates and to monitor their changing concentrations. We obtained the first evidence of a complex sexual communication system that actively counters the male chemical mate-guarding through use of an anti-antiaphrodisiac, to produce an honest indicator of female readiness to mate.

## Results

### Persistence of antiaphrodisiac effect

Mated females are less likely to be courted by a male than virgin females on the first day after mating ($\chi^2$ = 19.309, df = 1, p<0.001). However, this effect only persists for a few days and does not appear to influence the behavior of all potential mates (*Figure 1*). Over time the rate of courtship increases so that by five days after mating those females are as likely to be courted as virgins ($\chi^2$ = 0.004, df = 1, p=0.953). The variability and gradual increase in courtship observed may be the effect of changing levels of antiaphrodisiac being emitted by the female.

### Odorant components of spermatophore

GC-MS was used to identify and quantify three compounds that are found in the male accessory glands and in spermatophores dissected from females at various dates after mating (an example chromatogram of a spermatophore measured two days after mating is provided in *Figure 2*). We confirmed the presence of the previously identified antiaphrodisiac (*Brent and Byers, 2011*) myristyl acetate (MA), and also found two diterpenes, geranylgeranyl acetate (GGA) and geranylgeraniol (GGOH). The first two are found in high concentrations (tens of ng) in the male accessory glands, but GGOH is observed there only in trace quantities (*Figure 3*). When the spermatophore is sampled shortly after mating, the concentration of GGOH begins to increase, peaking two days later and subsequently declining. MA and GGA decline over this same period but the decrease in GGA is much more precipitous, likely as a result of the combined effects of externalization and conversion to GGOH. GC-MS analysis of the headspace of mated females over the same post-mating span indicated that these compounds were externalized in concentrations proportional to the amount found in the spermatophores, although by four days after mating the concentrations were too low to be accurately quantified (*Figure 4*).

### Male responses to spermatophore components

Male responses to the three compounds were tested using synthetic versions (500 ng in 1 µl of ethanol) applied topically to virgin females. The results varied significantly across the treatments ($\chi^2$ = 25.993, df = 7, p<0.001). Males exhibited less interest in courting MA-treated females relative to controls (*Figure 5*; 2 × 2 $\chi^2$ = 5.762, df = 1, p=0.016). The male response to MA appears to be dosage dependent (*Figure 6*; $\chi^2$ = 37.466, df = 4, p<0.001). Males courted treated females as often as controls when the topical application concentration was $5 \times 10^{-10}$ g µl$^{-1}$ ($\chi^2$ = 0.004, df = 1, p=0.953), but any amount of MA above that elicited a similarly significant suppression of courtship (p<0.01) suggesting an activation threshold rather than a gradient response (*Figure 6*). Unlike MA, applications of either GGA ($\chi^2$ = 0.686, df = 1, p=0.407) or GGOH ($\chi^2$ = 0.041, df = 1, p=0.839) at the same dosage did not significantly change courtship behavior compared to ethanol (*Figure 5*). Courtship rates did not change when GGA was combined with MA ($\chi^2$ = 0.018, df = 1, p=0.894) or GGOH ($\chi^2$ = 0.041, df = 1, p=0.840) relative to individual applications of these compounds. In

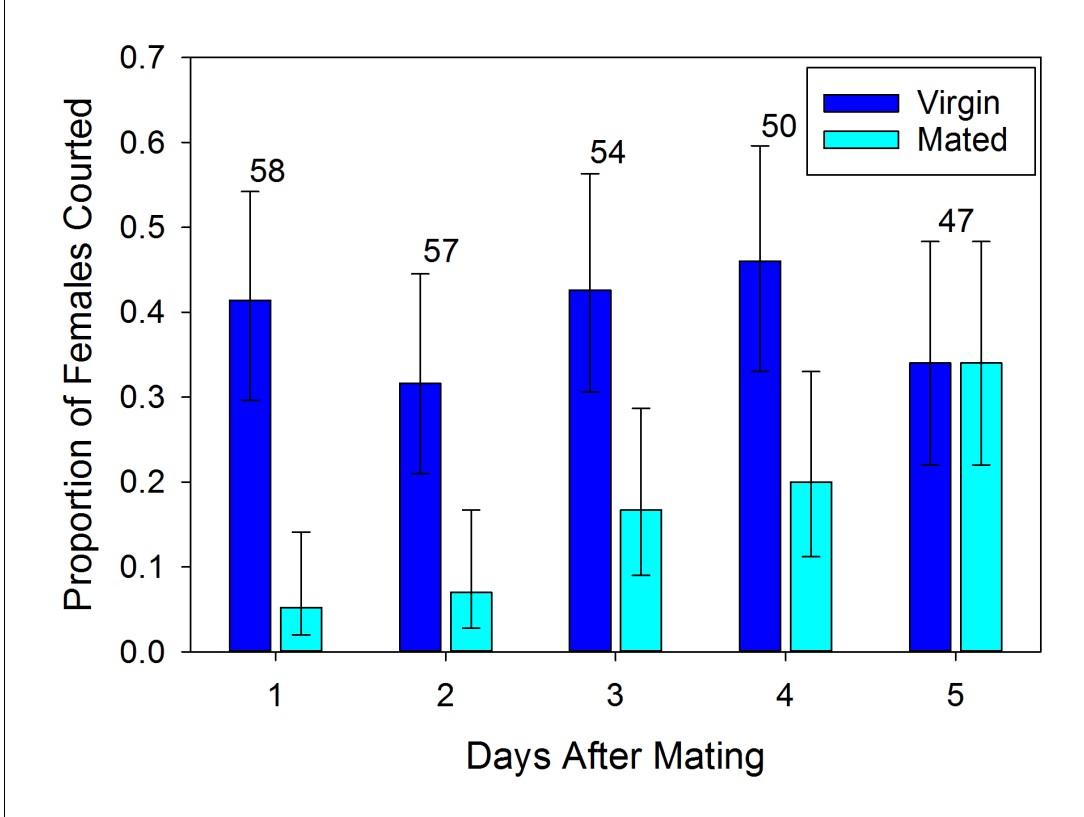

**Figure 1.** Proportion (±95% binomial confidence limits) of recently mated *L. hesperus* females courted by virgin males at different intervals after insemination relative to courtship rates for similarly aged virgin females. Numbers above bars indicate sample sizes. Rates of courtship differs significantly between mated and virgin females for each of the initial four days (2 × 2 $\chi^2$-tests, p<0.05).

contrast, combining GGOH with MA counteracted the antiaphrodisiac effect of the latter so that courtship rates were as high as the control ($\chi^2$ = 0.168, df = 1, p=0.682). Topical application of 1 or 10 ng GGOH to newly mated females was also able to overcome the natural antiaphrodisiac effect of the full suite of seminal constituents (*Figure 7*; $\chi^2$ = 16.506, df = 3, p<0.001). However, applying GGOH in combination with both MA and GGA did not prevent females from becoming as unattractive to males as females treated with just MA (*Figure 5*; $\chi^2$ = 6.994, df = 1, p=0.008), potentially indicating a synergistic effect of GGA with MA to reduce female attractiveness.

## Discussion

Mating causes a precipitous decline in the attractiveness of an *L. hesperus* female (*Strong et al., 1970*), but over the course of several days she becomes increasingly likely to be courted by males (*Brent, 2010a*). Our previous investigation of this phenomenon led us to conclude that when a male inseminates a female, he transfers an antiaphrodisiac that is slowly released through the female's gonopore and dissuades other males from courting her, and that the gradual dissipation of this antiaphrodisiac restores the female's attractiveness (*Brent and Byers, 2011*). This simple picture was consistent with the conclusions of previous antiaphrodisiac studies (*Andersson et al., 2000*, *Andersson et al., 2003*; *Schulz et al., 2008*; *Forsberg and Wiklund, 1989*; *Carlson and Langley, 1986*; *Schlechter-Helas et al., 2011*), however the results of our current investigation indicate a more complicated signaling system regulating *L. hesperus* reproductive behavior. Three compounds appear to be involved: myristyl acetate, the previously identified antiaphrodisiac (*Brent and Byers, 2011*), geranylgeranyl acetate and geranylgeraniol. The final compound, rather than being transferred directly from the male like the first two, is primarily produced within the female via chemical

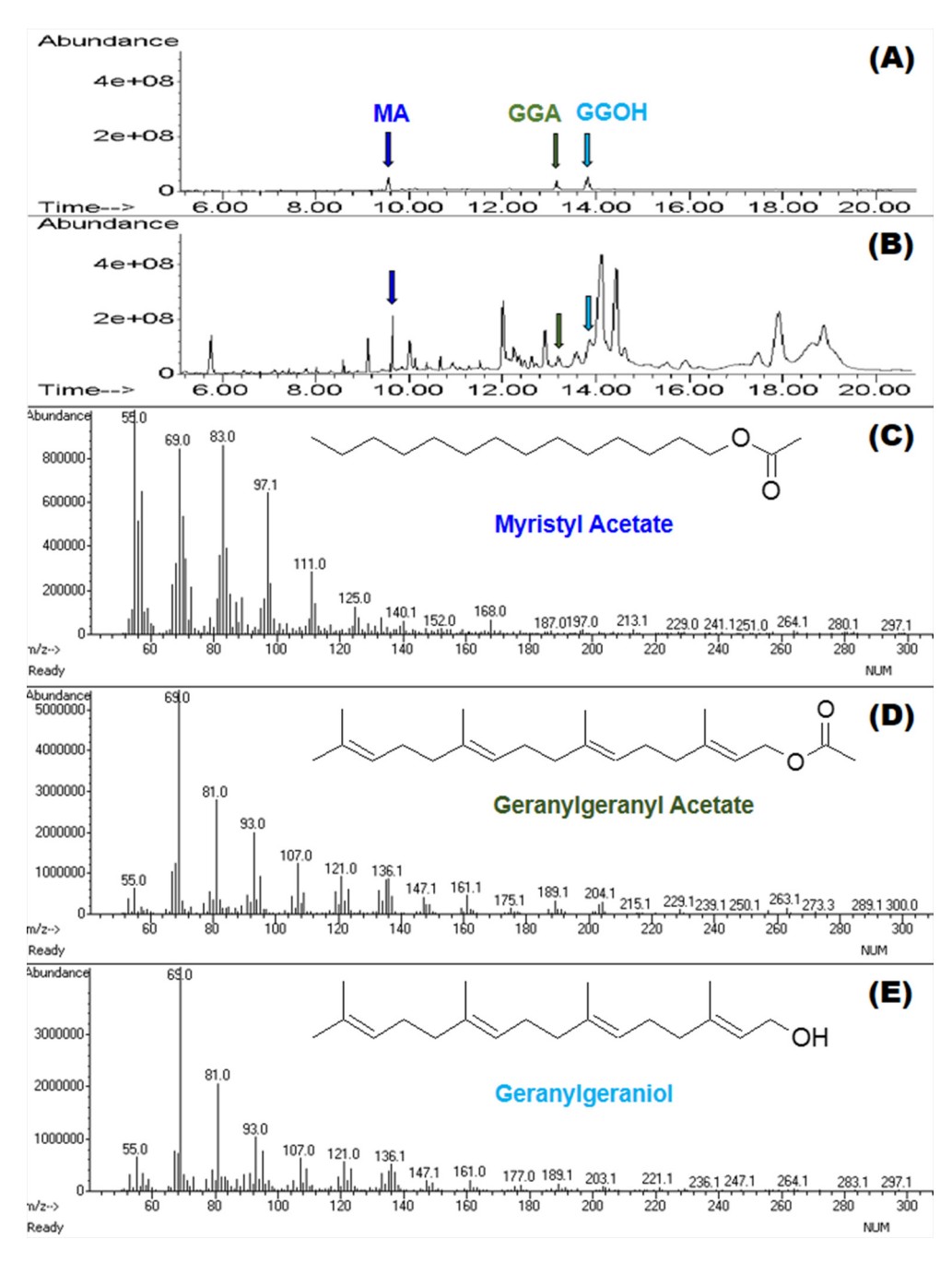

**Figure 2.** MSD ion chromatograms on GC column of (**A**) the three standards, and (**B**) a hexane extract of five pooled spermatophores excised from females two days after mating. Highlighted by arrows are peaks with matching retention times for myristyl acetate (MA), geranylgeranyl acetate (GGA) and geranylgeraniol (GGOH). Also shown are the mass spectra and chemical structures for each of the three compounds: (**C**) MA, (**D**) GGA, and (**E**) GGOH.

conversion of the GGA. While MA's role has been reconfirmed here, GGOH appears to play a counteracting role, diminishing or negating the effect of MA. In no instance did application of GGOH result in courtship rates being elevated above that for solvent controls, indicating that the

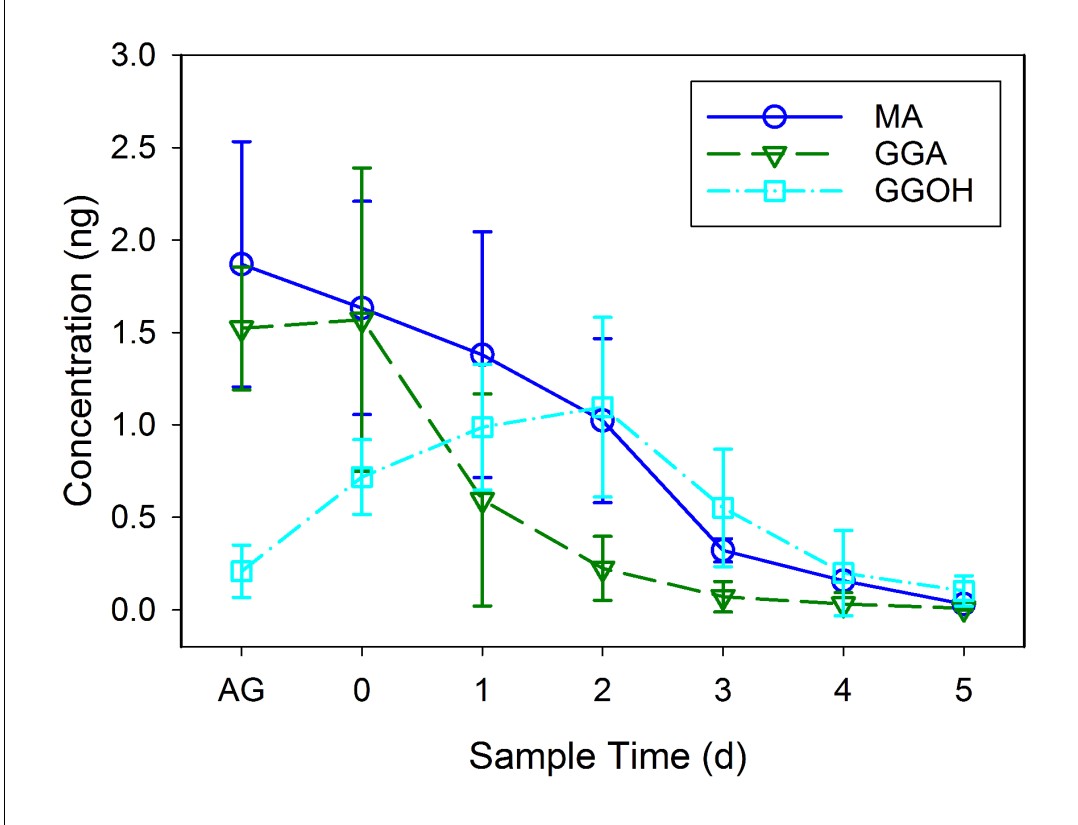

**Figure 3.** Mean concentration (± SD) of myristyl acetate (MA), geranylgeranyl acetate (GGA) and geranylgeraniol (GGOH) per individual sets of accessory glands (AG) of virgin males and in individual spermatophores from females sampled at daily intervals after mating. GC-MS was used to quantify ten samples consisting of five pooled spermatophores for each interval.

The following source data is available for figure 3:

**Source data 1.** Contents of virgin male accessory glands and spermatophore taken from mated females either shortly after insemination or after a 1–5 day long interval.

compound does not act as an attractant or excitatory agent. The only observable effect was to counteract the antiaphrodisiac effect, thus GGOH appears to serve as an anti-antiaphrodisiac.

The signaling function of GGA is not as easily categorized. When applied by itself, GGA did not significantly affect male courtship rates, although some depression is visible. GGA also did not make females less likely to be courted when combined with MA. Of course the dose of MA used in the mixture may have been sufficient to achieve the maximal level of repellency to males by itself, leaving little room for an additive effect to be observed. However, when added to a mix of MA and GGOH, GGA was able to counteract the anti-antiaphrodisiac effect of GGOH. The resulting reduction in female attractiveness suggests that GGA contributes towards the antiaphrodisiac effect, but only in a context-specific manner. The depletion of GGA over time will diminish this repellant effect, and when combined with the commensurate increase in GGOH, will accelerate the process of the female again becoming attractive.

Compared to our previously hypothesized single-compound signal communication system of *L. hesperus*, the greater complexity we have found could facilitate much better coordination between male courtship and the cessation of the female refractory period. During this three to seven day span subsequent to insemination, a female refrains from interaction with courting males (*Brent, 2010a*), a behavioral transition largely driven by male seminal products (*Brent, 2010a*; *Brent et al., 2016*). Because the female may copulate several times throughout her life (*Strong et al., 1970*; *Brent et al., 2011*), she cannot afford to have prospective mates repelled by

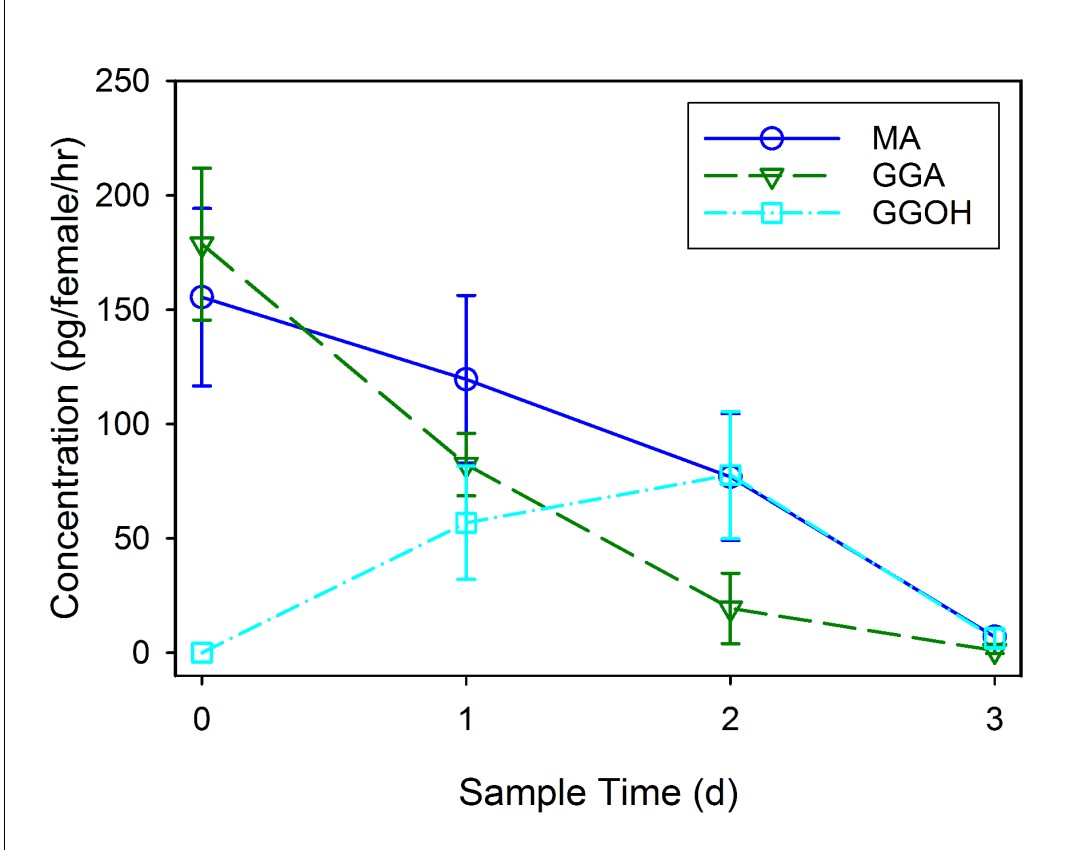

**Figure 4.** Mean concentration (± SD) of externalized myristyl acetate (MA), geranylgeranyl acetate (GGA) and geranylgeraniol (GGOH) found in the headspace of four groups each comprised of 20 females sampled at daily intervals after mating. Data is shown for calculated individual female emittance.

The following source data is available for figure 4:

**Source data 1.** Headspace measures of four groups of 20 mated females sampled by SPME for 2 hr at 24 hr intervals, starting on the day of mating.

an overly persistent antiaphrodisiac. For a certain period after mating an antiaphrodisiac would be an honest indicator of a female's receptivity and would help prevent males from engaging in unnecessary sperm competition (*Estrada et al., 2011*; *Malouines, 2016*). However, given that the amount of MA that a male transfers can be highly variable, the signal would not be a reliable indicator of a female's suitability and readiness to mate over the natural refractory period. Like females of *Pieris napi* (*Andersson et al., 2004*), there is no indication that *L. hesperus* females can modulate the rate of antiaphrodisiac release, so they are unlikely to have control over how quickly the supply can be depleted. Thus, if the initial MA concentration in a spermatophore is quite high, such as when contributed by a male that has not mated for a week or more, the female would be rendered unattractive well past when she again becomes reproductively receptive such that both sexes lose out on potential mating opportunities. If the initial MA is quite low, such as when contributed by a male that had already mated within the past day, a still refractory female would be harassed by unwanted suitors.

By utilizing additional signaling elements in this reproductive communication system, males can more accurately estimate the likelihood that a female is suitable to mate again, an added capability that would have been selectively favored (*Arak and Enquist, 1995*; *Estrada et al., 2011*). During the first couple of days after mating when male courtship is inhibited (*Brent, 2010b*), the combined emissions of MA and GGA are greater than GGOH. Once there is sufficient GGOH produced to offset the antiaphrodisiac effect, subsequent males will try to court a mated female. The starting

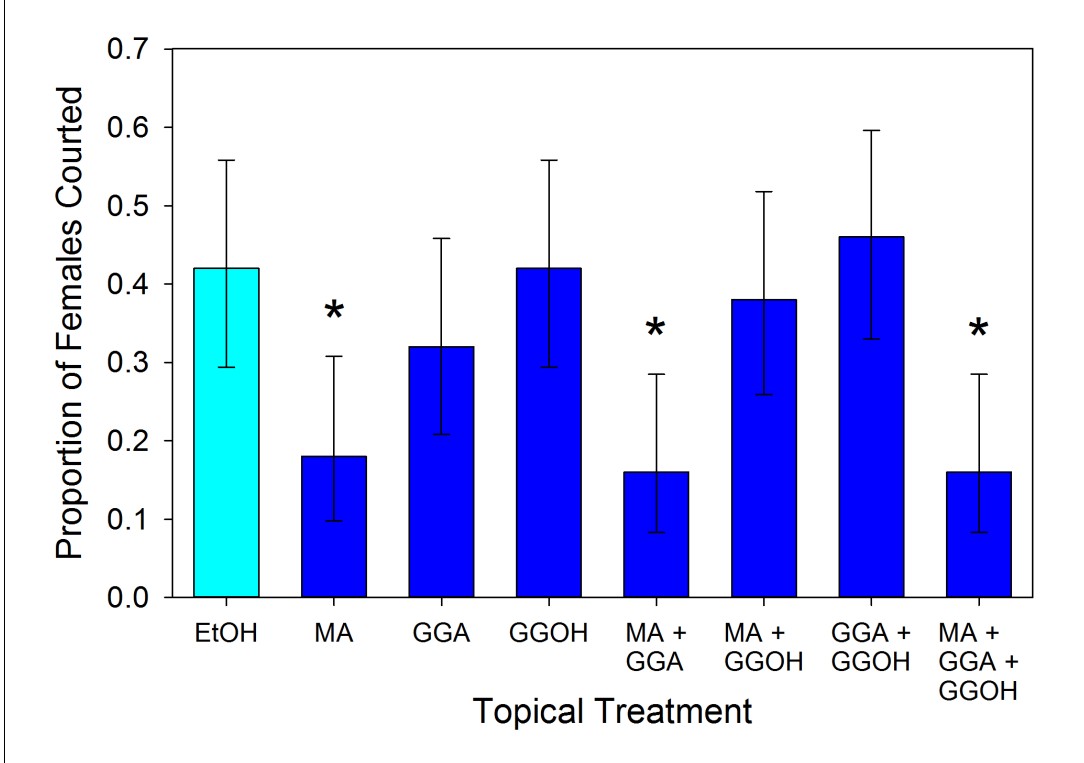

**Figure 5.** Proportion (±95% binomial confidence limits) of virgin females courted by virgin males after the females were topically treated with myristyl acetate (MA), geranylgeranyl acetate (GGA), geranylgeraniol (GGOH), or a combination of the compounds (5 ng compound in 1 µl of ethanol), and with ethanol as control. Treatments with an asterisk over them differed significantly in frequency from the EtOH control (2 × 2 $\chi^2$-tests, p<0.05). For all treatments, n = 50.

concentration of MA and GGA can vary, so long as they are above activation thresholds needed to repel other males. The relative quantities of MA and GGA transferred from a male are consistently similar although absolute amounts can vary between males. The amount of GGOH ultimately produced within the female is intrinsically tied to the starting quantity of GGA. Thus the signaling system is not affected by the variability between males in the quantities of pheromones transferred because only the time-dependent ratios are what matter. A high starting concentration of GGA will eventually produce a commensurately high concentration of GGOH to offset a high starting level of MA. So long as the conversion of GGA to GGOH occurs at a predictable rate, males will be able to accurately determine when a female is ending her refractory period.

It is likely that a combination of absolute quantities and relative proportions of these chemicals determine female attractiveness. This is suggested by some apparent inconsistency in the data. By the second day after mating, the concentration of GGOH appears comparable to that of MA, and GGA is substantially reduced. Despite these early shifts in proportions, the courtship rates remain suppressed until the fifth day after mating. GGA at this late stage may have fallen to such a low level that it no longer has a synergistic effect with MA to counteract the GGOH. We show here that MA has an activation threshold to be effective, but after that relative amounts of GGA and GGOH may be key modulators. Relative amounts of other, as yet unidentified, minor seminal constituents might also influence female attractiveness. Another consideration is that the threshold concentration identified for MA, and activation concentrations of the other compounds, may be substantially lower than our tests indicate. The external application of the chemical compounds does not fully mimic their normal release from the female gonopore. Applied chemicals can rapidly degrade, volatilize, or bind with chemicals on the cuticle. This would quickly diminish the perceivable amounts of the regulatory pheromones during the course of the bioassay, so that females avoided by males at the start were attractive again by the end. With each compound having different chemical properties, the effects

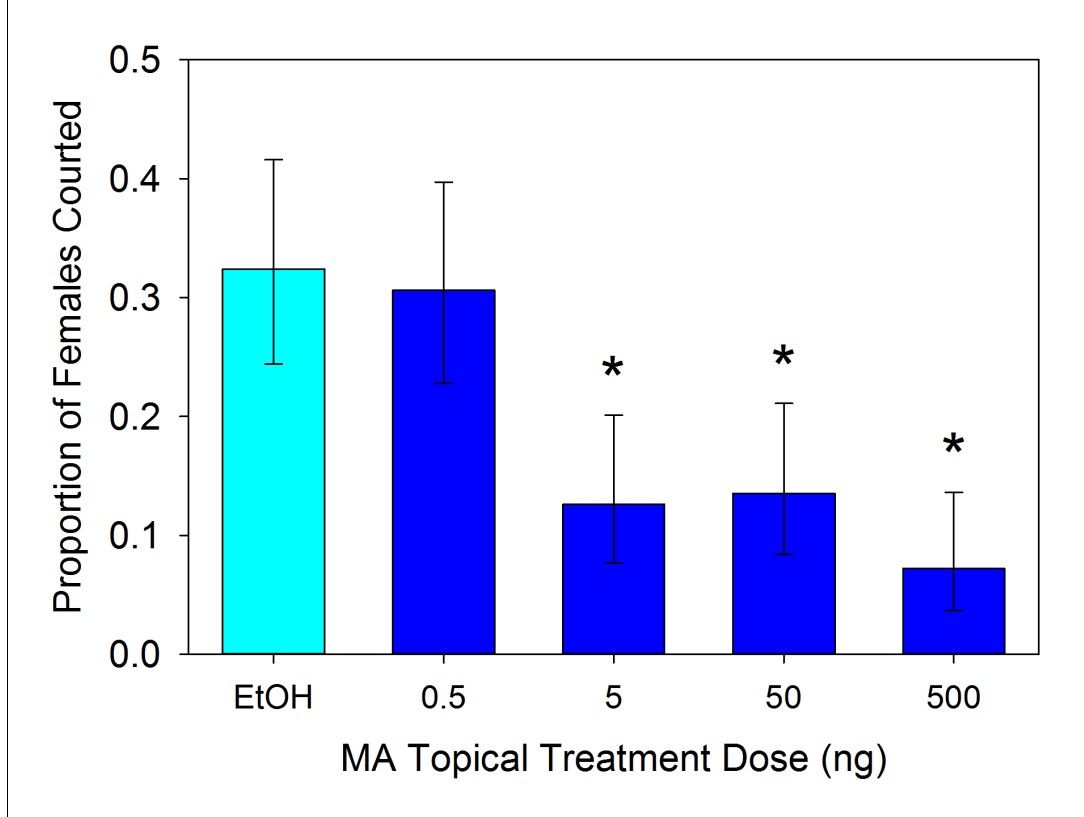

**Figure 6.** Proportion (±95% binomial confidence limits) of virgin females courted by males after the females were topically treated with one of four concentrations of myristyl acetate (MA) in ethanol or ethanol alone (n = 111 each). An asterisk indicates a treatment that evoked significantly fewer courtships than the ethanol control (2 × 2 $\chi^2$-tests, p<0.05).

of external application will not necessarily be consistent in these tests. Additional assays are needed to sort out the contributions toward female attractiveness of concentrations and proportions.

Another noteworthy finding was the variability in the responsiveness of males to these compounds. Even under the highest topical dosages of MA some males did not stop courting. This may be the result of diminished odorant receptor expression in males that cannot detect the antiaphrodisiac. Alternatively, a subgroup of males may employ a different mating strategy from the norm and are willing to tolerate sperm competition by indiscriminately courting every female encountered regardless of indicators of recent mating. This can be periodically rewarding as they are likely to be the first to access a female coming out of her refractory period earlier than her release of MA would otherwise indicate. We have also found a small but consistent subset of females that fail to enter into a refractory period, mating multiple times within a short span. Males willing to mate with one of these promiscuous females may benefit by fertilizing at least a portion of her eggs.

Although this is the first recorded instance of a pheromone being utilized as an anti-antiaphrodisiac, such compounds may not be rare. Just as it is likely that the occurrence of antiaphrodisiacs is far more widespread than has been reported due to a lack of thorough investigations (*Malouines, 2016*), anti-antiaphrodisiacs may have been overlooked until now because no one knew to search for them. Given the advantages of a two-dimensional chemical signaling system, it is quite likely that this is not the only instance. In fact, the evolution of a controlled antiaphrodisiac expulsion mechanism in *Drosophila* females (*Laturney and Billeter, 2016*), supports the likely occurrence of other such active counter measures to chemical mate guarding.

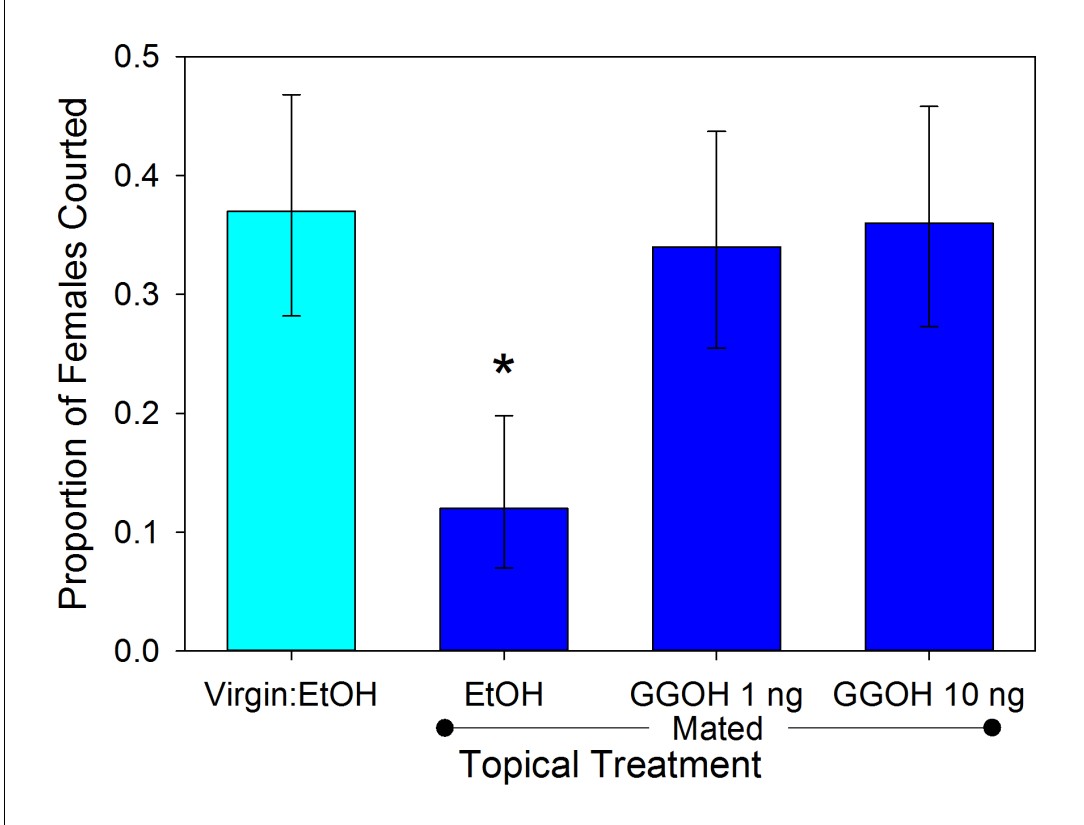

**Figure 7.** Proportion (±95% binomial confidence limits) of similarly aged females courted by virgin males. The females were virgins or newly mated and treated topically with ethanol, or mated and treated with 1 or 10 ng GGOH in ethanol (n = 100 each). Only ethanol treated mated females were courted less often than control virgins ($\chi^2$ =12.654, df = 1, p<0.001), indicating that GGOH can counter the full antiaphrodisiac blend of a mated female at biologically relevant doses.

## Materials and methods

### Insects

*L. hesperus* were obtained from an established colony at the USDA-ARS Arid Land Agricultural Research Center (Maricopa, AZ, USA). Colony health and genetic diversity were maintained by periodic outbreeding with locally-caught conspecifics. Insects were reared at 27°C, 20% relative humidity, under a 14:10 hr (L:D) photoperiod. Adults were produced from groups of mixed-sex nymphs reared in waxed paper containers covered with nylon mesh to ensure adequate ventilation and light exposure. Each container was provisioned with approximately 20 g of fresh green bean pods (*Phaseolus vulgaris* L.) and 12 g of artificial diet (*Debolt, 1982*) packaged in Parafilm M (Pechiney Plastic Packaging, Chicago, IL, USA) (*Patana, 1982*). Provisions were replaced every 48 hr. Daily monitoring allowed adults to be collected within 24 hr of emergence. Cohorts of same-aged adults were separated by gender and reared under conditions similar to those for nymphs, with the exception that the artificial diet was replaced with raw sunflower seeds (*Helianthus annuus* L.).

### Temporal degradation of antiaphrodisiac effect

A no-choice behavioral assay was used to test male courtship response to females after different time intervals subsequent to their being mated. All males were aged 7–9 days post-eclosion to ensure they were sexually mature and willing to copulate (*Brent, 2010a*). They were isolated from the opposite gender throughout adulthood to prevent their behavior towards prospective mates from being influenced by post-mating refractoriness (*Strong et al., 1970*; *Brent, 2010b*), and to guarantee the males were naïve with regard to the odor associated with mated and unreceptive

females (*Brent, 2010b*). A large cohort of 7d old virgin females were housed with similarly aged virgin males and allowed to mate for six hours. A group of 300 mated females were selected from this pool based on the presence of a visible spermatophore just below the surface of the cuticle (*Cooper, 2012*). The attractiveness of these mated females to males was determined at 1, 2, 3, 4 or 5 days after mating. Due to mortality over the test period, daily sample sizes ranged from 47 to 58. Females and males were only used once. During the interval between mating and testing, females were held in isolation in clean $1.5 \times 5.0$ cm covered glass Petri dishes with a 1 inch section of green bean and two sunflower seeds, which were changed out every other day. To determine post-mating attractiveness, a male was introduced to a female's dish and the pair was observed for 1 hr during which all instances of courtship were recorded. Courtships were distinguished from incidental approaches by characteristic male behaviors indicating the intent to mate (*Strong et al., 1970*). Females were dissected after each trial to ensure they had previously mated, as indicated by the presence of a spermatophore (*Brent, 2010b*).

## GC-MS of spermatophore components

The identification of myristyl acetate in the spermatophore using GC-MS was previously described (*Brent and Byers, 2011*). The same approach was used to identify geranylgeraniol and geranylgeranyl acetate as components of the spermatophore. GGA and GGOH identities were confirmed by additional comparisons of retention times and mass spectra of male Lygus accessory gland extracts and authentic standards.

To determine if there are temporal changes to male-derived pheromonal components after delivery into the female, spermatophore composition was analyzed every 24 hr for 5 days after mating. To produce the spermatophores, adult female and male *L. hesperus*, both aged 7 days, were mated together. Females were then housed in individual Petri dishes, as described above, then dissected at different intervals to remove the seminal depository and the spermatophore. The initial spermatophore sampling occurred within three hours of mating. Subsequent samples were taken every 22–25 hr thereafter. For each sample, five spermatophores were pooled. For each time tested, there were a total of 10 samples. Accessory glands, the source of the pheromones (*Brent and Byers, 2011*), were also sampled from virgin 7 day old males, again pooling five per sample and analyzing 15 samples. Tissues were homogenized twice in 200 µl of hexane (Sigma Aldrich, St. Louis, MO, USA) in a conical glass vial (Wheaton Scientific Products, Millville, NJ, USA) using a glass rod, then centrifuged at 1200 rcf. The resultant supernatant was stored in a clean glass vial with a teflon lined cap at $-80°C$ until analysis. Just prior to analysis the supernatant was dried down under nitrogen to an injection volume of 3 µl.

Samples were analyzed using a 7890A Series GC (Agilent Technologies, Sanata Clara, CA, USA) equipped with a 30 m x 0.25 mm Zebron ZB-WAX column (Phenomenex, Torrance, CA, USA) coupled to a 5975 C inert mass selective detector (Agilent Technologies). The helium carrier gas was programmed for constant flow of 1.2 ml min$^{-1}$. Samples were manually injected into the GC port at 250°C using the splitless mode. The oven/column temperature was initially held at 60°C for 1 min, and then increased at a rate of 20 °C min$^{-1}$ to 240°C, where it was held for 35 min. Samples were analyzed using the MS SIM mode using set m/z values for MA (55, 69, 83, 97), GGA and GGOH (69, 81, 93, 107). Total abundance was quantified against standard curves for each compound. The detection limit of the assay is approximately 0.5 pg.

## Syntheses

Hydroxyl groups of 1-tetradecanol (Acros Organics, Geel, Belgium), and geranylgeraniol (Toronto Research Chemicals, Toronto, Canada) were acetylated by acetic anhydride and pyridine to obtain the corresponding acetates (*Zada et al., 2004*). Myristyl acetate was purified by vacuum distillation using glass oven Kugelrohr (Buchi, Flawil, Switzerland) at 115°C/1 mm, 90% yield, >98% purity. Geranylgeranyl acetate was purified by column chromatography ($SiO_2$/ether:hexane 95:5) 80% yield, 96% purity. Purity was determined by GC (Agilent 6890) with FID detector.

## Head space measurements

Head space samples were collected from four groups of mated females. For each group, thirty females, aged 8–10 days, were mated to virgin males of the same age over a three hour period.

Female mating status was confirmed by observation of the spermatophore through the abdominal cuticle (*Cooper, 2012*). For each test, 20 females were randomly selected from those mated and were still alive on the day of data collection. Females were placed in a 50 mL volumetric flask stoppered with a rubber septum through which was inserted a solid-phase microextraction fiber (SPME) with an 85 μm Polyacrylate coating (Sigma). Prior to collection, the SPME fibers were preconditioned by heating within the GC injector (250°C) for one hour. Sample collection occurred over a two hour period, either within three hours of mating (Day 0) or every 24 hr thereafter through four days. Samples were immediately analyzed as described above by direct insertion of the SPME into the injection port of the GC-MS. Fibers were kept in the injector port throughout the run. Mating status of the sampled females was subsequently confirmed by dissection. A total of four samples were run for each time period using the same groups of mated females.

## Pheromone effects on courtship

The same behavioral assay as described for testing the attractiveness of mated females was used to test for a dose-dependent effect of myristyl acetate on male courtship. Virgin 7 day old females were topically treated with 1 μl of ethanol (95%) containing myristyl acetate diluted at each of four concentrations (0.5, 5, 50 and 500 ng μl$^{-1}$), or 1 μl of an ethanol control. Males were introduced to the dishes for one hour and instances of courtship recorded. A total of 111 trials were conducted during which all five treatments were run concurrently in separate dishes. Dishes were cleaned between trials.

The same approach was used to determine male courtship responses to virgin female treated topically with the pheromones either individually or in combination. Treatments had 500 ng of each compound in 1 μl of ethanol in the following mixtures: MA, GGA, and GGOH alone, MA with GGA, MA with GGOH, GGA with GGOH, all three combined, and ethanol as control. Fifty trials were run in which the eight treatments were run concurrently using the same cohort of animals.

The efficacy of GGOH as an anti-antiaphrodisiac was also demonstrated by topical treatments and recording male responses. As above, males were exposed to virgin females treated with ethanol, or to newly mated females treated with ethanol or GGOH at concentration of 1 or 10 ng μl$^{-1}$. A total of 100 trials using females of the same cohort were conducted with all four treatments run concurrently in separate dishes.

## Statistical analysis

The effect of the topical applications on the attractiveness of females to males was determined by n x 2 $\chi$2-tests comparing the proportion of males exhibiting courtship behavior across all treatments and when significance was indicated individual comparisons between treatments were conducted using 2 × 2 $\chi$2-tests. To reduce the incidence of false negatives, comparisons were made only between treatments and the virgin female control group, not between all treatments. The Dunn-Šidák correction was used to adjust the significance level for multiple comparisons. Analyses were conducted using Sigmaplot 13.0 (Systat Software, Point Richmond, CA, USA). Overall, no data points were excluded as outliers. Biological but not technical replicates were used. Appropriate minimum sample sizes were determined by power analysis for multiple pairwise comparisons, using $\pi$ = 0.80 with both $\alpha$ and $\beta$ set to 0.05. Preliminary experiments provided anticipated rates of courtship for mated and virgin females at 15% and 45% respectively.

## Acknowledgements

We thank Dan Langhorst, LeAnn Elhoff (ARS) and Daniela Fefer (ARO) for their assistance with behavioral and chemical assays and syntheses. Mention of trade names or commercial products in this article is solely for the purpose of providing specific information and does not imply recommendation or endorsement by the U.S. Department of Agriculture. USDA is an equal opportunity provider and employer.

## Additional information

### Funding
There was no outside funding for this work.

### Author contributions
CSB, Conceptualization, Resources, Data curation, Formal analysis, Investigation, Methodology, Writing—original draft, Project administration; JAB, Conceptualization, Resources, Investigation, Methodology, Writing—review and editing; AL-Z, Resources, Writing—review and editing

### Author ORCIDs
Colin S Brent, http://orcid.org/0000-0003-2078-1417
John A Byers, http://orcid.org/0000-0002-7233-6334

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
