## [Decision Letter]

Thank you for submitting your article "An insect anti-antiaphrodisiac" for consideration by *eLife*. Your article has been reviewed by three peer reviewers, one of whom, Leslie C Griffith, is a member of our Board of Reviewing Editors and the evaluation has been overseen by a Senior Editor. One of the other two reviewers, Claude Everaerts, has also agreed to share his identity.

The reviewers have discussed the reviews with one another and the Reviewing Editor has drafted this decision to help you prepare a revised submission.

This is an interesting and generally well done study on a new and interesting class of insect pheromones: anti-antiaphrodesiacs. Using the plant bug Lygus hesperus the authors show that females convert a male-derived compound (geranyl geranyl acetate; GGA), that is transferred during mating, into geranylgeraniol (GGOH). Using a combination of behavioral and chemical analysis, the authors conclude that female-produced GGOH counteracts the anti-antiaphrodisiac effects of myristyl acetate (MA), another male-derived compound.

While the finding is interesting and novel, there are a couple issues of quantification and analysis that need to be cleared up before the study is ready for publication.

1) The chemical profile measured from mated females does not completely correlate with the loss of female attractiveness.

Figure 1 shows that mated female attractiveness is still suppressed at 4 days post-mating. However, Figure 3 reveals that by day 4, MA, GGA, and GGOH from the spermatophore transferred to females are each below 0.5 ng. According to Figure 6, MA is no longer effective as an anti-aphrodisiac at 0.5 ng. Also, according to Figure 5, equimolar amounts of GGOH to MA are sufficient to counter the anti-aphrodisiac effects of MA. By Day 2 post-mating, GGOH and MA levels in the spermatophore are approximately equivalent yet female attractiveness is still reduced.

While MA contributes to the post-mating female loss of attraction, it may not be the only factor. Similarly, the conversion of GGA to GGOH may have anti-antiaphrodisiac effects but this process alone is not sufficient for restoring female attractiveness. Some discussion of the fact that not all the effects can be accounted for is required.

2) It is interesting that the addition of GGOH is sufficient to counteract mating-induced loss of attractiveness in newly mated females (Figure 7). However, females are treated with 500 ng of GGOH. Yet, only approx. 1-1.5 ng is found in the spermatophores from mated females. The amount used for treatment is excessive and makes it difficult to interpret the results. In order for these experiments to be meaningful, the authors need to show a dose-dependency of GGOH and its anti-antiaphrodisiac effects that is within biologically meaningful levels.

3) Statistical analysis could be improved: (i) why are authors using 2x2 Khi square, and not nx2 Khi square with post-hoc determination of significance?; (ii) why not compar the temporal evolution of the MA, GGA and GGOH concentrations and correlate them with behavioural data?

---

## [Author Response]

*While the finding is interesting and novel, there are a couple issues of quantification and analysis that need to be cleared up before the study is ready for publication.*

*1) The chemical profile measured from mated females does not completely correlate with the loss of female attractiveness.*

*Figure 1 shows that mated female attractiveness is still suppressed at 4 days post-mating. However, Figure 3 reveals that by day 4, MA, GGA, and GGOH from the spermatophore transferred to females are each below 0.5 ng. According to Figure 6, MA is no longer effective as an anti-aphrodisiac at 0.5 ng. Also, according to Figure 5, equimolar amounts of GGOH to MA are sufficient to counter the anti-aphrodisiac effects of MA. By Day 2 post-mating, GGOH and MA levels in the spermatophore are approximately equivalent yet female attractiveness is still reduced.*

*While MA contributes to the post-mating female loss of attraction, it may not be the only factor. Similarly, the conversion of GGA to GGOH may have anti-antiaphrodisiac effects but this process alone is not sufficient for restoring female attractiveness. Some discussion of the fact that not all the effects can be accounted for is required.*

We agree that what we present may not be the entire mechanism involved in regulating female attractiveness and that our data is not completely consistent. We more fully address these issues within the Discussion. We point out that a large part of the perceived discrepancies are likely due to limitations of the available methodology. While the amounts extracted from females (Figure 3) are accurate in relative amounts, the measures may underestimate the real amounts present in living females because of possible destruction/isomerization and binding of the three compounds when extracted with other oils/fats and chemicals in insect body tissues. This is a potential problem in all studies of chemical extraction from insects. Additionally, male sensitivity to the pheromones is likely to have been underestimated. Although our results suggest that between 0.5 and 5 ng of MA is needed to cause an anti-aphrodisiac effect (Figure 6), the activation concentration might be considerably lower. Application of the putative pheromones to the external body does not fully mimic the natural situation of release from the female’s gonopore. Externally applied chemicals can be rapidly degraded or become bound, leading to an exponential decline in release rate over the span of the bioassay. Further, each chemical will respond differently on the cuticle, causing increasing discrepancies in relative concentrations during the bioassay.

We agree that MA may not be the only factor in post-mating loss of female attraction, but the preponderance of evidence in this and our previous study point to MA as the primary determining factor. We provide some evidence that GGA may contribute to loss of female attraction, and this may be significant at the natural release rates and ratios which are difficult to replicate in bioassays. The loss of MA and GGA certainly contributes in large part to restoring female attractiveness. The conversion of GGA to GGOH within the female would accelerate the process of removing GGA (contributing to restoring female attractiveness) as well as producing the GGOH that would also contribute to restoring female attractiveness. Other processes in the female that removed MA would also contribute to restoring female attractiveness. These issues are now covered more thoroughly in the Discussion

*2) It is interesting that the addition of GGOH is sufficient to counteract mating-induced loss of attractiveness in newly mated females (Figure 7). However, females are treated with 500 ng of GGOH. Yet, only approx. 1-1.5 ng is found in the spermatophores from mated females. The amount used for treatment is excessive and makes it difficult to interpret the results. In order for these experiments to be meaningful, the authors need to show a dose-dependency of GGOH and its anti-antiaphrodisiac effects that is within biologically meaningful levels.*

We agree that a test of dose-dependency of GGOH on its anti-anti-aphrodisiac effects would be useful for a better understanding. The original dosage of 500 ng was chosen because the compounds could be degraded or bound by cuticular chemicals. It is also a typical quantity compared with many other semiochemicals. We merely wanted to be sure to counter the effects of MA and GGA in newly mated females. However to ensure the acceptability of the results, we have repeated this part of the experiment using topical applications of GGOH on newly mated females with the more biologically relevant dosages of 1 and 10 ng. Even at these lower concentrations, the GGOH was still found to be effective at blocking the anti-aphrodisiac properties of the seminal constituents of newly mated females (Figure 7).

*3) Statistical analysis could be improved: (i) why are authors using 2x2 Khi square, and not nx2 Khi square with post-hoc determination of significance?; (ii) why not compar the temporal evolution of the MA, GGA and GGOH concentrations and correlate them with behavioural data?*

We have clarified our Materials and methods to reflect that n x 2 chi-square analyses were performed for data in Figures 5 to7 to first determine whether there were significant effects on the proportions of females being courted. Once significance was indicated, individual 2 x 2 analyses were performed, with post-hoc adjustments (Dunn-Šidák correction) for multiple tests. For the data in Figure 1, each daily test was independent, thus the 2 x 2 chi-square analysis by day was directly applied.

Direct comparisons of the temporal changes in pheromone concentration and behavior was not done because these kinds of data were collected under very different (essentially incompatible) conditions using different sets of animals. Release of these compound is exceedingly difficult to measure because of their volatility and absorption by SPME or other collection methods. Extraction can determine the ratios and amounts reasonably well (except if there is differential component absorption to tissue components) but the comparison in bioassays is again difficult because of the chemical absorption by the cuticle and the exponential decline in release rate of each component and the widely differing volatilities of MA compared to GGA and GGOH. We do weigh the likely link in the temporal information presented in Figure 1, Figure 3 and Figure 4 during the Discussion.